# Migration of a Retained Surgical Suture Needle in the Common Bile Duct

**DOI:** 10.3390/diagnostics12102276

**Published:** 2022-09-21

**Authors:** Tzu-Cheng Wen, Kuo-Hua Lin, Yang-Yuan Chen

**Affiliations:** 1Department of Gastroenterology, Endoscopy Center, Changhua Christian Hospital, Changhua 500, Taiwan; 2Department of Gastroenterology, Changhua Christian Hospital, Changhua 500, Taiwan

**Keywords:** retained surgical foreign bodies, suture needle, common bile duct

## Abstract

Retained surgical foreign bodies have been a cause of concern since physicians began operating on patients. Retained surgical foreign bodies in the common bile duct (CBD) are rare and may cause cholangitis and jaundice. We report the case of a patient who initially presented with fever and right upper-quadrant abdominal pain. He had received cholecystectomy and choledochojejunostomy 28 years ago and had been well since then. Abdominal computed tomography (CT) revealed left-lobe liver abscess and a linear curve of high-density material. Endoscopic retrograde cholangiopancreatography (ERCP) displayed mild dilatation of the common bile duct (CBD) and choledojejunostomic fistula of the middle CBD. A curved, linear, rusty, metallic surgical suture needle was detected and successfully removed under ERCP.

##  

**Figure 1 diagnostics-12-02276-f001:**
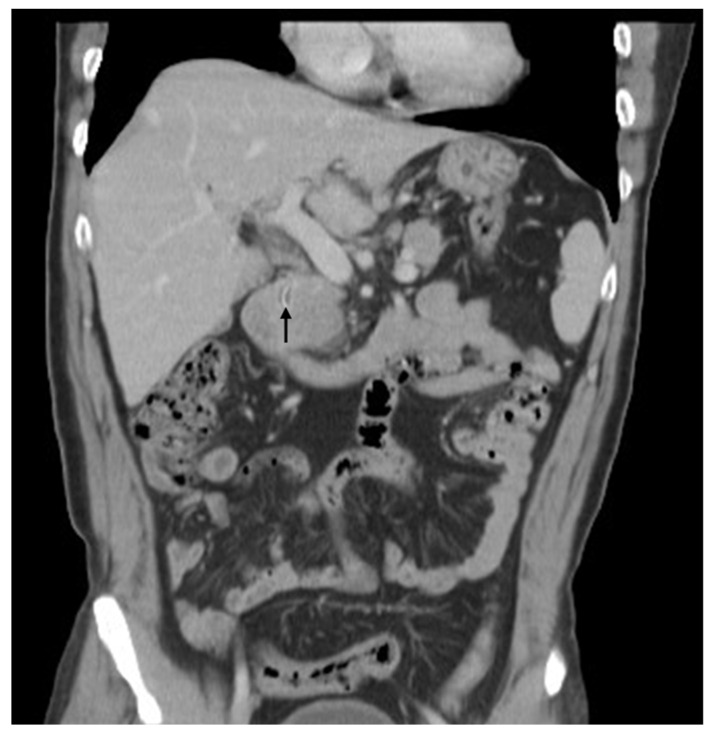
A computed tomography (CT) scan of the abdomen demonstrated abscess over the left lobe of the liver and a linear curve of high-density material within a mass. A 58-year-old man presented at our hospital with right upper-quadrant pain and fever. He had undergone laparotomic cholecystectomy and choledochojejunostomy 28 years prior at another hospital. He had been healthy until 3 days before his admission, when his symptoms appeared. A computed tomography (CT) scan of the abdomen demonstrated abscess over the left lobe of the liver and a linear curve of high-density material within a mass (Figure 1; arrow). The patient received antibiotics and underwent aspiration of liver abscess. Endoscopic retrograde cholangiopancreatography (ERCP) was performed after the symptoms and signs had subsided. ERCP depicted mild dilatation of the CBD and choledojejunostomic fistula of the middle CBD. The patient underwent a biopsy forceps, which removed a 1.7 cm curved, linear, rusty, metallic surgical suture needle containing bile (Figure 2). We examined the patient’s abdominal radiographs, which revealed that the needle was on the right side of the third lumbar spine vertebra (Figure 3; arrow). We followed up with abdominal radiography and detected no further evidence of the needle. The patient was discharged without further events.

**Figure 2 diagnostics-12-02276-f002:**
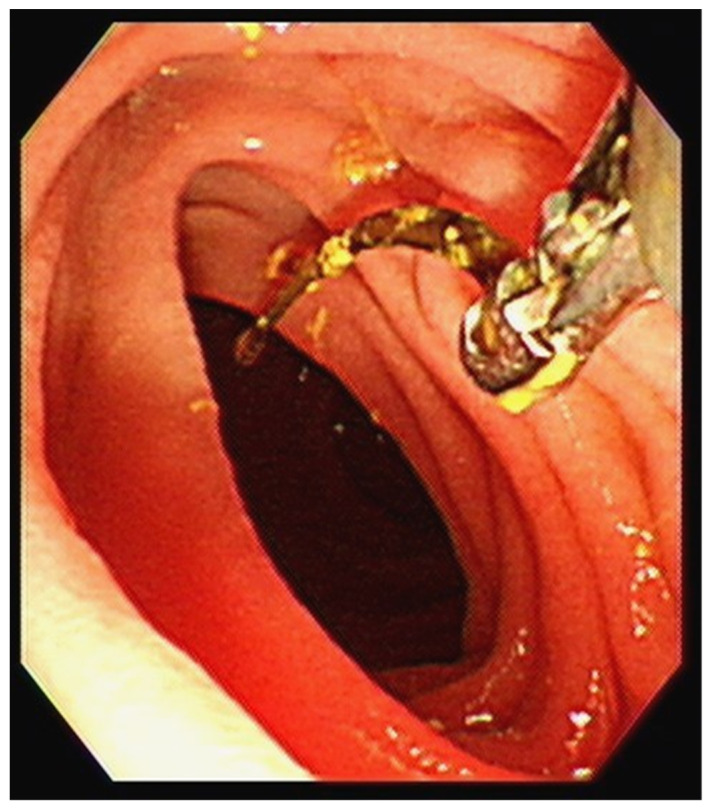
The patient underwent a biopsy, which revealed a 1.7 cm curved, linear, rusty, metallic surgical suture needle containing bile. A retained surgical item poses a hazard to surgical safety and occurs occasionally despite various systems and safeguards [1,2,3]. Any surgical material, including tools, supplies, and equipment, can inadvertently be left inside a patient’s abdomen and can cause subsequent harm to the patient. Several factors increase the risk of the retention of surgical foreign bodies including patient characteristics such as obesity, case-specific conditions such as emergencies, complex surgical procedures, involvement of more than one surgical team, unplanned changes in the procedure, prolonged surgical procedures, and operating room’s culture and environment [4]. Intraoperative and postoperative imaging may assist in the rapid identification of the site of foreign bodies, as it did with our patient’s retained needle [5]. Although radiography easily detects retained surgical needles longer than 1 cm, the detection of smaller needles less than 1 cm in length is more difficult [6]. Compared with plain radiography, abdominal CT can provide more accurate findings.

**Figure 3 diagnostics-12-02276-f003:**
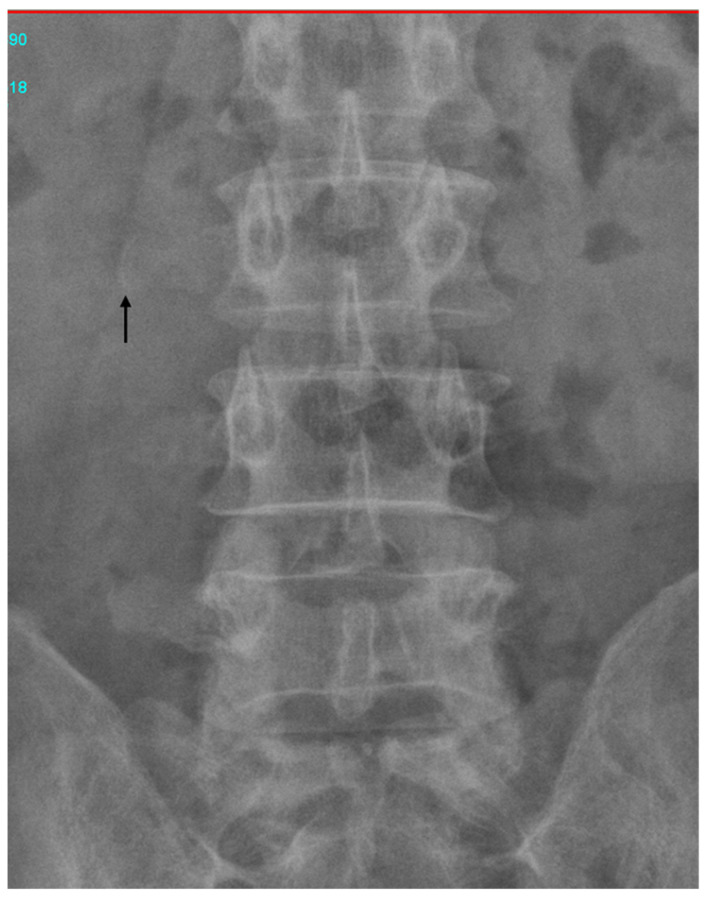
Abdominal radiographs, which revealed that the needle was on the right side of the third lumbar spine vertebra. The symptoms and signs of a retained surgical item include adhesion, foreign body migration, visceral perforation, and abscess formation [1]. These symptoms can occur early in the postoperative period or may develop after months or years [5]. In prior instances, retained surgical suture needles were surgically removed. However, our patient developed symptoms and signs of a retained needle 28 years after his surgery, by which point the needle had migrated to the ampulla of vater, from where it was easily removed through ERCP. Migrated surgical clips, stents, gauze pieces, suture materials, and fragments of t-tubes have all been reported as retained iatrogenic foreign bodies causing CBD obstruction and subsequent sequelae [2]. To our knowledge, this is the first reported case in which a retained surgical needle migrated to the ampulla of vater. Needles comprise 0.06–0.11% of foreign bodies that are retained during surgery [7]. Preventing instances of unintentionally retained surgical materials is a critical problem. A nationwide Brazilian study of retained surgical foreign bodies reported that challenging medical situations, security protocol omission, and inadequate work conditions contributed to retained foreign bodies. Sponges are notoriously overlooked because they are routinely inserted into cavities to expose the operative field. Thus, a preventive protocol could involve the introduction of the use of sponge-holding forceps [8]. In conclusion, CBD obstruction caused by foreign bodies can be safely ameliorated through ERCP without complications noted during follow-up.

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
