# Peer review of "Migration of a Retained Surgical Suture Needle in the Common Bile Duct"

_diagnostics, 2022, doi:10.3390/diagnostics12102276_

Round 1

Reviewer 1 Report

This is a clinical case report on migration of a retained surgical needle in bile duct.  The uniqueness of the report is that needle was left in the abdomen for 28 years until the patient was admitted to the hospital due to right upper-quadrant pain and fever. The report overall is well written and highlighted the needs to design innovative surgical equipment to minimize the chance to retain medical instruments in patients’ body post-surgery.

Minor comments: English spelling needs to be checked.

For example: Line 53: Figure 2 Legend: Should be “The patient”; Line 56: Figure 3 Legend:  Should be “Abdominal”; Line 80: Should be “sequelae”

Author Response

point to point response:

Reviewer 1:

  1. We change he patient to The patient.
  2. We change abdominal to Abdominal.
  3. We change sequala to sequelae.

Thank you very much for your reviewer. If you have any questions

regarding this manuscript, please do not hesitate to contact us.

Reviewer 2 Report

An interesting case report.

The following suggestions are offered to improve the manuscript:

1. An image of cholangiogram should be given to demonstrate the findings mentioned in the text.

2. The biopsy part in the retrieval of needle is not understood. Please clarify 

Author Response

point to point response:

Reviewer 2:

  1. A computed tomography (CT) scan of the abdomen can easily see metallic surgical suture needle within distal common bile duct.
  2. We change to “The patient underwent a biopsy forceps, which removed a 1.7-cm curved, linear, rusty, metallic surgical suture needle containing bile.”

 Thank you very much for your reviewer. If you have any questions

regarding this manuscript, please do not hesitate to contact us.
